# Prediction of Time Series Gene Expression and Structural Analysis of Gene Regulatory Networks Using Recurrent Neural Networks

**DOI:** 10.3390/e24020141

**Published:** 2022-01-18

**Authors:** Michele Monti, Jonathan Fiorentino, Edoardo Milanetti, Giorgio Gosti, Gian Gaetano Tartaglia

**Affiliations:** 1RNA System Biology Lab, Department of Neuroscience and Brain Technologies, Istituto Italiano di Tecnologia, Via Morego 30, 16163 Genoa, Italy; 2Centre for Genomic Regulation (CRG), The Barcelona Institute for Science and Technology, Dr. Aiguader 88, 08003 Barcelona, Spain; 3Center for Life Nano- & Neuro-Science, Istituto Italiano di Tecnologia, Viale Regina Elena 291, 00161 Rome, Italy; jonathan.fiorentino@helmholtz-muenchen.de (J.F.); edoardo.milanetti@uniroma1.it (E.M.); Giorgio.Gosti@iit.it (G.G.); 4Department of Physics, Sapienza University of Rome, 00185 Rome, Italy; 5Department of Biology and Biotechnology Charles Darwin, Sapienza University of Rome, 00185 Rome, Italy

**Keywords:** recurrent neural network, dual attention, gene regulatory network, time series prediction

## Abstract

Methods for time series prediction and classification of gene regulatory networks (GRNs) from gene expression data have been treated separately so far. The recent emergence of attention-based recurrent neural network (RNN) models boosted the interpretability of RNN parameters, making them appealing for the understanding of gene interactions. In this work, we generated synthetic time series gene expression data from a range of archetypal GRNs and we relied on a dual attention RNN to predict the gene temporal dynamics. We show that the prediction is extremely accurate for GRNs with different architectures. Next, we focused on the attention mechanism of the RNN and, using tools from graph theory, we found that its graph properties allow one to hierarchically distinguish different architectures of the GRN. We show that the GRN responded differently to the addition of noise in the prediction by the RNN and we related the noise response to the analysis of the attention mechanism. In conclusion, this work provides a way to understand and exploit the attention mechanism of RNNs and it paves the way to RNN-based methods for time series prediction and inference of GRNs from gene expression data.

## 1. Introduction

Recent technological innovations, such as chromatin immunoprecipitation sequencing (ChIP-seq) and RNA sequencing (RNA-seq), allow complex networks formed by interactions among proteins, DNA and RNA to be systematic studied [1]. These approaches are strongly advancing system biology and open up new opportunities in medicine, allowing us to study complex diseases that affect several genes [2,3].

The complex interaction network among genes in a cell forms the gene regulatory network (GRN). In a GRN, there are, typically, transcription factors (TFs) proteins and target genes. TFs activate and inhibit the transcription of target genes. In turn, target genes produce other TFs or proteins that regulate cell metabolism [4]. The recurrent architectures constituting the building blocks of GRNs, known as network motifs, have been widely studied and classified in the last decade through systems biology approaches [5,6,7]. In parallel, mathematical models, including both logical and continuous models based on ODEs and information theory, allowed the properties of GRNs to be inferred in different contexts [8,9,10,11]. The recent emergence of high-throughput experimental techniques for gene expression measurement led to the development of a wide range of computational methods to obtain an accurate picture of the interactions among TFs and their target genes, which is known as the GRN inference problem [12,13,14,15]. Deep learning models have been recently employed to understand gene regulation [16,17] or to infer GRNs from imaging [18] and RNA-seq gene expression data [19,20]. For this reason, we investigated how deep neural networks (DNN) can be exploited to classify GRNs considering different network topologies.

Predicting the behavior of a stochastic network is an extremely challenging task in modern science. Indeed, the measurable output of many processes, such as the evolution of the stock market, meteorology, transport and biological systems, consists of stochastic time traces [21,22,23,24,25]. Inferring information on the behavior of these systems gives the possibility to predict their future changes, in turn allowing one to have a thorough knowledge of the evolution of the system and, on that basis, make functional decisions. Many mathematical methods relying on inference theory have been developed, which, often, are meant to learn the mean properties of the systems and guess the future behavior of the various system-specific variables [26]. Knowing the actual future behavior of a stochastic system allows one to infer parameters of a mathematical model that is suited both to describe the system and to predict the evolution of data as well as possible. These models can be defined analytically, but the number of parameters to infer grows exponentially with the size of the network, making the system difficult or even impossible to learn [27].

Analytical models give the possibility of inferring the physical parameters of the system, or at least parameters that have a well-defined meaning for the system. This aspect plays a crucial role because it allows one to obtain a physical interpretation of the network due to the interaction between its constituent elements. In this sense, a learning process is not only able to estimate the prediction of temporal evolution data, but it also provides the generative model, which is, in some cases, physically interpretable. Somehow parallel to the general problem of predicting the dynamical evolution of a time sequence, there is the task of classifying a set of time series. In this context, the estimated parameters obtained from the predictive model can be used in a classifier to classify different time series [28]. Among several predictive methods based on machine learning techniques, neural networks (NNs) have undergone extensive development in recent years. In particular, DNNs are powerful mathematical architectures that are well suited to perform tasks where ordinary machine learning procedures on a given analytical model fail or cannot be used [29]. However, the parameters of DNNs do not have a straightforward physical interpretation, meaning that the model is able to predict the dynamic of the system, but it does not provide direct information about the variables. DNNs perform a set of vector–matrix and matrix–matrix linear operations, concatenated via other linear or non-linear transformations, in order to give a certain output given an input vector. One of the key points in using DNNs is the process of setting parameters, which depends both on learning the network from the input data and on the selected architecture. Among the various types of neural networks, recurrent neural networks (RNNs) are best suited for the prediction of time series data [30]. Basically, these networks have a layer that is used to feed the network at each time step together with the input vector, conferring a sort of memory to the network. In this sense, it is possible to shape the network architecture relying on the theoretical analysis of the power of predicting time-varying signals taking information from the past. The amount of information from the past in a model, as well as how it is processed, depends on the type of problem we have to solve. For instance, in order to predict the next time step of a time sequence of a noisy process, we can think that we do not need information from far in the past, as it could be redundant and misleading.

In this work, a DNN approach was adopted with the aim of inferring information on the behavior and on the structure of GRNs. To evaluate the effectiveness of this approach, we generated a synthetic gene regulatory model and its stochastic evolution in time and we used a deep neural network to predict the future behavior of the time traces of the system. Through a series of statistical analyses of the results obtained, we took into account the DNN parameters in order to infer information on the physical structure of the gene interactions. We show that this approach can be used to categorize protein expression profiles obtained from GRNs with different topological structures, e.g., master regulator genes or oscillating gene networks.

At the heart of our approach, there is the consideration that high-throughput experimental techniques, such as microarrays and RNA-seq, can produce time traces of gene expression. For instance, whole-genome microarrays have been employed to monitor the gene expression dynamics of circadian genes in cyanobacteria, providing the expression levels of approximately 1500 genes over 60 h, measured at intervals of 4 h [31,32]. Another work used metabolic RNA labeling and a comparative dynamic transcriptome analysis (cDTA) to measure the expression of 479 genes in synchronized budding yeast cells at 41 time points during three cell cycle periods [33]. Moreover, single-cell techniques, such as single-cell reverse transcription quantitative polymerase chain reaction (scRT-qPCR) and single-cell RNA-sequencing, can provide temporal gene expression measurements, as witnessed by studies probing gene expression at different time points [34,35] or relying on pseudo temporal ordering of single cells when a single time point is measured [36,37,38]. Recently, a single-cell transcriptomic technique able to preserve cell viability, called Live-seq, has been developed, making the application of our method to experimental data even more promising [39]. However, these experimental techniques do not provide any information on the interactions among genes [40]. Knowing the structure of the underlying gene regulatory network is crucial to understand how the system would respond to perturbations, such as mutations or new external inputs. This is particularly relevant for the study of complex diseases, for which the genes involved are known. Among all, various types of cancer are characterized and described with this approach, since they often originate from a modification or an alteration of the gene regulatory network that governs a given cellular task [40]. Therefore, inferring the physical structure of a GRN from the time behavior of its output is a key problem in biology and the ability of having control over could open the doors to a broader understanding of the biological mechanisms at the origin of pathologies, thus suggesting new strategies from a diagnostic and therapeutic point of view.

In this work, we used the dual attention mechanism (AM) that has been recently introduced [41]. Using this approach, it is possible to accurately predict the next time step of a time series relying on the previous time points. The idea is that, from an endogenous set of variables evolving in time, it is possible to predict the behavior of a target variable. More formally, in a stochastic network that evolves in time, to predict the state of the i-th variable, the previous *T* time steps of the ensemble of variables that are likely responsible for its evolution need to be considered. Moreover, the dual attention mechanism is needed to infer the functional interaction among the variables. The first AM selects the genes out of the pool that are master regulators of the given state. The second AM is applied to the selected genes of the first level and it prioritizes important time steps of their expression. These two attention mechanisms are integrated within an LSTM-based recurrent neural network (RNN) and are jointly trained using standard backpropagation. In this way, the dual attention recurrent neural network (DA-RNN) adaptively selects the most important time steps of the most relevant inputs and captures the long-term temporal dependencies. Since our goal is to predict the dynamic of all the genes that belong to the GRN, we built a scheme parallel to the DA-RNN. We use RNNs because we know that these neural networks can store several patterns and that the network structure affects the dynamics and the number of the stored patterns [42,43,44].

Regarding the forecasting task, the DA-RNN has been recently shown to outperform state-of-the-art methods for time series prediction and different types of RNNs without the dual attention mechanism [45,46].

To evaluate the general usefulness of the DA-RNN, we generated synthetic time series of gene expression from different classes of GRNs, resembling known biological networks. We trained a parallel DA-RNN for each GRN and we showed that it predicted the future behavior of the time traces with high accuracy. Next, we focused on the input attention mechanism, with the goal of gathering information about the structure of the different GRNs used. Relying on graph theory and network analyses, we studied different properties of the attention layer, finding that, in general, they were able to discriminate different GRN architectures, thus identifying a structural information about the GRN in the attention mechanism of the DA-RNN. We observed that the robustness of the prediction by the DA-RNN to noise addition was different for the various GRN architectures and we compared it to the properties of the input attention. Our work represents a first promising application of a DA-RNN to the prediction of time series gene expression data and it provides a step beyond in the analysis of the interpretability of the attention mechanism of deep neural networks.

## 2. Materials and Methods

### 2.1. Dual Attention Mechanism Structure and Learning Procedures

Instead of using a classical neural network approach, we propose a new method to predict the concentrations of a set of proteins.The method uses state-of-the art deep learning methodologies, including long short-term memory (LSTM) and attention mechanism (AM).

We implemented a deep neural network model that relies on the dual-stage attention-based recurrent neural network (DA-RNN) for time series prediction [45], which has been developed in order to predict the behavior of a target quantity in a stochastic ensemble of time traces. The network has an encoder–decoder structure and, in both parts, an attention layer is the central core (Figure 1A). The encoder attention layer weights the *N* time traces to optimize the prediction of the target, while the decoder attention layer looks for the best way of weighting the past time points in each time trace. The combination of these two layers finds two key characteristics of the prediction for a given target time trace; on one hand, it finds the most important elements (in our case, genes) in the system that should be considered for predicting the target and, on the other hand, it quantifies the importance of the different past points.

We are interested in predicting the further time step of the whole network of interacting genes. To this end, we implemented the parallel scheme shown in Figure 1B. Given the input data, *N* parallel DA-RNNs were trained, where the *i*-th network predicts the behavior of the *i*-th target gene. We hope to find, encoded in the attention layer, functional information about the interaction among the genes.

To optimize network performance, a standard scaling of the data was performed, converting all the time traces to zero mean and unit variance.

We used a mini-batch stochastic gradient descent (SGD) together with the Adam optimizer [47] to train the model. The size of the mini-batch was 128. The cost function of backpropagation is the mean squared error (MSE), as follows:(1)$$MSE\left(y_{p},y_{c}\right)=\frac{1}{M}\sum_{i=1}^{M}{\left(y_{p}^{\left(i\right)}\left(t\right)-y_{c}^{\left(i\right)}\left(t\right)\right)}^{2}\;\;,$$
where $y_{p}$ and $y_{c}$ are the vectors of the prediction and the target data, respectively, and *M* is their length.

#### Choice of the Time Window for Network Training

Regarding the training of the DA-RNN, we used one time series with 1001 time points (time from 0 to 10,000, sampled every 10, extremes included) for each gene regulatory network. To choose the time window for training the DA-RNN, we performed an analysis of the autocorrelation time of the gene expression time traces and an evaluation of the prediction performance of the DA-RNN. We looked at the distribution of the autocorrelation times for two example gene regulatory networks (details are provided in Section 3.2) and we noticed that, in the case of a fully connected network, it peaked around T=50 time points (Figure 1C). The autocorrelation times were selected as the first time lag for which the autocorrelation function entered the 95% confidence interval (computed using the Python function “acf” in the “statsmodels” package); an example is shown in Appendix C. Then, we tested several values of *T* for training the parallel DA-RNN and we computed the following: (i) the training time for all the genes of the GRN (Figure 1D); (ii) the root-mean-square error (RMSE) of the prediction (Figure 1E); (iii) the Kullback–Leibler divergence and the Jensen–Shannon distance between the probability distribution of the real data and that of the predicted ones (Figure 1F). To use the probability distributions for the quantification of the network performance, we evolved the prediction using time windows of length *T* time points. From Figure 1E,F, we can observe that the time window *T* did not significantly affect the prediction performance and the capability to correctly retrieve the full probability function of the real data, while Figure 1D shows that T=50 time points was still a suitable value from a computational point of view, taking approximately two hours to train a parallel DA-RNN with 20 genes on a single core.

We implemented the network in python using the py-torch library.

### 2.2. Deep Neural Network Parameter Analysis

To extract the weights of the input attention mechanism for each GRN, we trained a DA-RNN for each target gene, as described in the previous section, and we collected the vector of *N* input attention weights, where *N* is the number of genes. This resulted in the definition of a matrix $A=\{a_{ij}\}$ for each GRN, where the element $a_{ij}$ is the input attention weight of gene *j* obtained by training the neural network on target gene *i*. Next, we treated the input attention matrix as a graph and we computed three local network properties, i.e., the clustering coefficient, the betweenness centrality and the hub score. We used the “transitivity” function, “betweenness” and “hubscore” functions of the “igraph” package of R (the Igraph Software Package for Complex Network Research; *InterJournal* 2006, *Complex Systems*) for the three descriptors, respectively. For each case, we considered a weighted graph. Specifically, the three network descriptors are defined as follows:

(i) The clustering coefficient is given by
(2)$$C_{i}=\frac{1}{s_{i}}\frac{1}{\left(k_{i}-1\right)}\sum_{j,h}\frac{\left(w_{ij}+w_{ih}\right)\left(a_{ij}a_{ih}a_{jh}\right)}{2}\;\;,$$
where $s_{i}$ is the strength parameter of vertex *i*, which is defined as the sum of all weighed links. The $a_{nm}$ are elements of the adjacency matrix (with both *n* and *m* being *i*, *j* and *h* indices). Finally, $k_{i}$ is the vertex degree (the number of links) and $w_{nm}$ are the weights.

(ii) The betweenness centrality is defined as follows:(3)$$b_{i}=\sum_{k\neq i,l\neq i}\frac{s_{kl}\left(i\right)}{s_{kl}}\;\;,$$
where $s_{kl}\left(i\right)$ is the number of shortest paths (*s*) that go from node *k* to node *l* passing through node *i* and $s_{kl}$ is the total number of shortest paths from node *k* to node *l*.

(iii) The Kleinberg’s hub centrality score, or hub score, of the vertices is defined as the principal eigenvector of $AA^{t}$, where A is the adjacency matrix of the graph.

The interpretation of the three network properties is provided in Section 3.3.

### 2.3. Response of the Prediction to Noise Addition

In this section, we describe the analysis of the response of the prediction by the DA-RNN to noise addition on the protein concentration. For each target gene of a GRN, we used the parameters of the DA-RNN network, trained as discussed in Section 2.1, to predict its level at time t+1 using the previously predicted *T* time steps (cf. Section 3.2 and Section 3.3 for details). With this procedure, we predicted the whole time series for a given gene and we computed the mean squared error, defined in Equation (Equation 1).

Then, we repeated the prediction but added a Gaussian noise, with zero mean and variance $\sigma^{2}\in \left[0.1,0.9\right]$, with steps of 0.1. The noise was added to the protein concentration values of the target gene in the time window *T* used for the prediction and we computed the MSE between the noisy prediction and the original time series. In the end, for each gene regulatory network, we obtained a matrix containing the MSE for each target gene and different values of the variance of the Gaussian. To study systematically if there was a different response to noise addition in the prediction of gene expression dynamics for different gene regulatory network architectures, we collected the vectors of the mean and variance of the MSE, computed over the different values of $\sigma^{2}$, in a matrix. For each network, the vector of the mean was ranked in increasing order and normalized to its maximum, while the variances were ranked according to the mean ranking and they were also normalized to the maximum. This was carried out because there is not a biological correspondence between the genes in the different gene regulatory networks and to retain the relationship between mean and variance of the MSE for the same gene in each network.

### 2.4. Clustering and Principal Component Analysis

Given a matrix representing a specific network property of the input attention mechanism or the response of the prediction to noise for all the GRNs used, we performed a clustering analysis and a principal component analysis (PCA) using the Python package scikits-learn.

Firstly, we scaled the data to zero mean and unit variance using the function “StandardScaler”. Next, we performed agglomerative hierarchical clustering on each matrix, with complete linkage and Euclidean distance, using the function “AgglomerativeClustering”. For each matrix, we chose the optimal number of clusters as that maximizing the average silhouette score. The silhouette score for a given sample is defined as
(4)$$s=\frac{D_{N}-D_{I}}{\max\left(D_{I},D_{N}\right)}\;\;,$$
where $D_{I}$ is the mean intra-cluster distance for the sample under consideration and $D_{N}$ is the distance between the sample and the nearest cluster that the sample is not a part of.

We also ran a PCA on the same matrix, using the “pca” function.

In order to compare the dendrograms obtained from the hierarchical clustering of the different matrices, we first converted them to Newick trees, which is a standard format for phylogenetic trees, then we used the R package “ape” to read the dendrograms and the R package “TreeDist” to compute the distance between each pair of dendrograms. In particular, we used the information-based generalized Robinson–Foulds distance [48]. The Robinson–Foulds distance counts the number of splits that occur in both trees. Its generalization measures the amount of information that the splits of two trees hold in common under an optimal matching. The optimal matching between splits is found by considering all the possible ways to pair splits between trees.

Finally, the comparison between the partitions obtained from the hierarchical clusterings was performed using the variation of information [49], which measures the information that is gained or lost from a clustering to another of the same dataset. Given two partitions C and $C^{\prime }$ of the same dataset, the variation of information between them is defined as
(5)$$VI\left(C,C^{\prime }\right)=H\left(C\right)+H\left(C^{\prime }\right)-2I\left(C,C^{\prime }\right)\;\;,$$
where H(·) is the entropy of a partition and $I(C,C^{\prime })$ is the mutual information between the partitions.

## 3. Results

### 3.1. Gene Regulatory Network Dynamic

Modeling and simulating a whole gene regulatory network is a fundamental challenge for computational biophysics. Indeed, the network of interacting genes includes many chemical reactions, with the respective rates and parameters that define a high-dimensional parameter space. Moreover, the activation of a gene in the classic framework happens via a transcription factor (TF) that, in order to bind the DNA, usually goes through a set of crucial chemical post-transcriptional modifications, which can drastically change the nature of the protein. Other physical dynamics, such as tuned delay, repression, the presence of multiple TFs and the cooperativity among them, further complicate the endeavor of modeling gene regulatory networks.

The simplest way of describing a gene regulatory network is to define the nature of the interactions, writing the probability $P(x_{i}^{+})$ of activating a gene *i* expressing a protein with a concentration $x_{i}$ between time *t* and t+dt as
(6)$$\begin{array}{cc} P\left(x_{i}^{+}\right)dt=\prod_{j}\left[J_{ij}\frac{\left(J_{ij}+1\right)}{2}k_{ij}\frac{x_{j}^{h_{ij}}\left(t+\tau_{ij}\right)}{x_{j}^{h_{ij}}\left(t+\tau_{ij}\right)+K_{ij}^{h_{ij}}}+\right.\\ \left.+J_{ij}\frac{\left(J_{ij}-1\right)}{2}{\tilde{k}}_{ij}\frac{K_{ij}^{h_{ij}}}{x_{j}^{h_{ij}}\left(t+\tau_{ij}\right)+K_{ij}^{h_{ij}}}\right] & \mathrm{if}\;\;\;J_{ij}\neq 0\;\;, \end{array}$$
where we introduce the following items:$J_{ij}$ is the interaction matrix among the genes *i* and *j*, which is 1 if gene *j* expresses a protein, at concentration $x_{j}$, that is a TF for gene *i*, −1 if it is a repressor and 0 if the two genes do not interact.$k_{ij}$ and ${\tilde{k}}_{ij}$ are the expression and repression rates for the gene *i* from the TF (repressor) *j*.$h_{ij}$ is the cooperativity index of the ij interaction.$K_{ij}$ is the dissociation constant of the reaction ij.$\tau_{ij}$ is the delay between the expression of the protein $x_{j}$ and the time when it is ready to activate (repress) the gene *i*. It allows one to account for the translation of mRNAs into proteins and it has been previously included in Gillespie simulations [50,51,52]. However, in the present study, we set it to zero for the sake of simplicity.

Notice that the term $J_{ij}\frac{\left(J_{ij}+1\right)}{2}$ is one if $J_{ij}=1$ and zero otherwise. While the term $J_{ij}\frac{\left(J_{ij}-1\right)}{2}$ is one if $J_{ij}=-1$ and zero otherwise.

We stress that the behavior of the gene regulatory network described in Equation (Equation 6) is intrinsically dependent on the temporal dynamic of the respective number of proteins $x_{i}\left(t\right)$ expressed. For this reason, the correct way to look at that is to model the protein dynamics that, with high fidelity, mirror the gene regulatory dynamics. Indeed, we can consider that the concentration of a protein still behaves as a stochastic process and the probability of increase is directly proportional to the probability of expressing the correspondent gene. For our purpose, we considered that a protein can be created through the expression of a gene following the stochastic rules defined in Equation (Equation 6). Moreover, we included a dilution effect on the protein concentration that led to a constant decay rate. This definition of the stochastic process that describes the behavior of creation and dilution of a protein completely defines our Gillespie simulations; details are provided in Appendix A. Other effects can be taken into account, as we mention in Appendix B, in which we provide the computation of the mean field equations of the system.

### 3.2. Recurrent Neural Network for Stochastic Time Traces Generated by Different Gene Regulatory Network Architectures

Different classes of gene regulatory networks model distinct biological systems. In this section, we describe the gene regulatory network architectures that we chose for the generation of time series data and for training the deep neural network model. Since we aim at understanding the regulatory interactions between genes in a gene regulatory network, we generated synthetic data for which we know the ground truth network. Firstly, we focused the analysis on the degree of connectivity, which is an important network feature. Indeed, some networks are extremely connected, meaning that the matrix $J_{ij}$ is dense, while others are sparser, i.e., the matrix $J_{ij}$ has more null values. Here, we introduce the parameters $n_{r}$ and $n_{a}$, which are the probabilities that an element of the $J_{ij}$ is −1 (inhibitory connection) or 1 (activatory connection), respectively. In other words, $n_{r}$ and $n_{a}$ multiplied by *N* (total number of genes) can be interpreted as the mean number of repressors and activators for a given gene. Using these parameters, we can tune the connectivity matrix density of the net. Indeed, if we analyze the extreme values of these parameters, for $n_{r},n_{a}\to 1$, the net is fully connected, while, for $n_{r},n_{a}\to 0$, the net is fully sparse.

Oscillations are another important property of GRNs. Therefore, we included, in our analysis, oscillatory networks in which there is a central core of genes that evolve in an oscillatory fashion. The simplest way to induce oscillations in a gene regulatory system is to introduce a feed-forward loop with some delay. To this end, we relied on the Goldbetter model [53], whereby a first gene activates the expression of a second gene, which, in turn, activates a third one that, finally, represses the first node. This procedure introduces a delay in the repression of the first node that starts the oscillation. It can be intuitively understood by thinking that gene 1 starts to rise for a certain time τ, while genes 2 and 3 are getting expressed; after this time, the concentration of the protein expressed by the third gene is high enough to induce the repression of gene 1, which, in turn, stops its expression. Consequently, the concentration of the protein goes down and the oscillation begins.

Another class of networks that we studied is that where all genes share incoming connections from a master regulator gene. In other words, we define a GRN in which the first gene *i* shares out edges with all other genes; in addition to these edges, we added random connections among the other gene pairs.

The last class of GRNs that we studied is that connected to an external oscillatory signal. In addition, in this case, the connectivity among the internal genes is random and quantified by the parameters $n_{r}$ and $n_{a}$ defined above. We studied how our prediction model performed and learned varying these parameters and network topology. The parameters used in the Gillespie simulations are listed in Appendix C.

To predict the temporal evolution of a GRN, the neural network is trained on a time window *T* of the input data, which plays a fundamental role in the training procedure, as described in detail in Section 2.1 of the Methods.

In our approach, we considered that the mutual information between the state of the system at time t−Δt and the state of the system at time *t* goes to 0 for Δt→∞ [54], also for oscillatory systems [55,56].
(7)$$I\left[x(t-\Delta t);x(t)\right]\to 0\;\;\;\;\;\mathrm{if}\;\;\;\Delta t\to \infty \;\;.$$Moreover, the usual way to weight the past is using the temporal autocorrelation function $C\left(t,\tau \right)={\langle x\left(t\right)x\left(t-\tau \right)\rangle }_{\tau }$. Indeed, if two time series are correlated, there is an information exchange between them, while the opposite is not necessarily true. Thus, we used the autocorrelation time as a proxy to set the time window for training. To avoid losing any useful information, we set the time window slightly larger than the maximum autocorrelation time of the system. This information drove us to estimating the time window for training the DA-RNN (Figure 1C).

We used the parallel DA-RNN described in detail in Section 2.1. After training, we predicted the time series for all the genes of each gene regulatory network. The accuracy of the performance can be appreciated from Table 1, which shows the mean and standard deviation of the root-mean-square error (RMSE) of the prediction for each gene regulatory network.

The results shown in Table 1 were obtained by predicting the time series with time windows *T* of the original data. The RMSE computed with these predicted value is the standard performance measure for the time series prediction task [45,46]. However, it only quantifies the capability of the neural network to predict the next time step, knowing the previous *T* values. To also account for the long-range performance of the parallel DA-RNN in time series prediction, we introduce an additional way of predicting the time series after training the DA-RNN. The prediction is propagated by providing, to the network, the first *T* input data from the time trace and letting the DNN to predict the next time step. When the first time point (after *T*) is predicted, the network has, as the input for the next prediction, the T−1 input data plus the first prediction point. Iterating this procedure, after *T* steps the system would rely its prediction only on previously predicted data, showing a complete autonomous behavior. Using this procedure to calculate the prediction, we found that the DA-RNN was able to reconstruct well the dynamics of the system, as observed in Figure 2, for both oscillatory and non-oscillatory dynamics. Further, in Figure 3, we show, for some example genes, how well the time traces and the probability distributions of protein concentration were reconstructed for different GRNs. It is important to notice that oscillatory dynamics, which have a well-characterized mean behavior, were easily learned by the neural network, showing a very good match between the stochastic simulated data and those generated by the neural network. Additional results on the performance of the DA-RNN for each gene regulatory network, comparing the two ways of performing time series prediction, are reported in Appendix C.

### 3.3. Studying the Input Attention Mechanism and the Response of the Prediction to Noise

The input attention mechanism of the DA-RNN assigned weights to the genes of the gene regulatory network under study, with the goal of prioritizing those that enabled the prediction of the dynamic of the target gene. Thus, we reasoned that the set of input attention weights for a given gene regulatory network could reflect, to some extent, the structure of the regulatory interactions, allowing us to distinguish different architectures of the gene regulatory networks. For each gene regulatory network, we extracted the input attention matrix $A=\{a_{ij}\}$, as described in detail in the Methods section.

A recent study showed that the input attention mechanism of the DA-RNN does not reflect the causal interactions between the variables composing the system [57]. This is also supported by the extremely small values of the correlation coefficient between the true interaction matrix and the input attention matrix of each gene regulatory network, reported in Table 2.

Thus, to investigate the organization of the interactions that make up the network architecture from the input attention matrices, we employed methods from graph theory. We chose three different local parameters, which were defined for each node of the network and provided different levels of description. The first is the clustering coefficient, which describes the topological/structural organization of the interactions involving a node and its first neighbors, meaning that the local relationship structure that each node has with its first neighbors is evaluated. Specifically, the clustering coefficient is defined on the number of triangulations of each node. Therefore, not only node *i* should be connected both with node *j* and node *k*, but nodes *j* and *k* should also be connected. As a second descriptor, we considered the betweenness centrality, which analyzes the centrality of the node with a higher level of complexity since it is not obtained only from the interactions with the first neighbors (such as the clustering coefficient) but it is defined in terms of the number of all the shortest paths connecting all node pairs that pass through the given node. Indeed, a node *i* with a large value of betweenness is a key node of the network (i.e., it is a node with high centrality), because many shortest paths connecting the pairs of nodes pass through it. The last descriptor, the hub score, is directly related to the intrinsic properties of the adjacency matrix, since it is defined as the principal eigenvector of the adjacency matrix of the graph. This means that it is able to capture mathematical properties of the network given by the optimization of its information contained in the corresponding adjacency matrix. For the mathematical definition of the three descriptors, see the Methods section.

To analyze the role of each network parameter for each type of network used, we needed to consider the values of the descriptors for each node (gene) of the graph in order, in this case, decreasing. The reason for this choice is due to the fact that there is no correspondence of a physical and/or biological nature between the genes of two different matrices; therefore, there is no a priori numbering of the genes. We only evaluated two matrices that were similar to each other if the two profiles of the analyzed network property were similar. This procedure allowed us to compactly describe each attention matrix with a single vector composed of a specific network property, node by node, in descending order.

Next, we studied how the time series prediction by the DA-RNN was affected by noise added to the gene expression, to see if different regulatory architectures reacted in distinct ways. The addition of noise to the time series was meant to reflect the alteration in genes belonging to specific networks coming from different extrinsic sources [58,59,60,61]. Moreover, modeling the effects of noise is relevant to understand variations in experimental techniques such as RNA-seq [62] and mass spectrometry [63].

To this end, we used the parameters of the DA-RNN, trained as described in the Methods section, to predict the level of the target gene, but we added a Gaussian noise with zero mean and variance $\sigma^{2}$ on the previous *T* predicted values of the protein concentration, where *T* is the time window used for training the DA-RNN. We computed the MSE on the prediction and we repeated the procedure for several values of $\sigma^{2}$. In the end, we built a matrix containing the mean and the variance of the MSE, computed over the different values of $\sigma^{2}$, for each gene and each GRN. We refer the reader to the Methods section for further details.

As a first comparison between the matrices obtained for each descriptor from graph theory and from the results of the noise addition, we represented each pair of matrices in scatter plots, shown in Figure 4, and we computed the Pearson correlation coefficient. There are other possible measures of similarity between these matrices, such as the mean squared error between pairs of elements, but, since we are interested in the general features of the structure of these matrices, we computed the Pearson correlation coefficient among pairs of matrices of the graph theory descriptors as a first measure of the structural similarity. To obtain a comparable shape, in the case of the noise analysis, we retained only the vectors of the mean MSE. We observed the presence of a high correlation between each pair of matrices, suggesting that they encoded the properties of the corresponding GRN similarly, an aspect that we explore further in the following sections. Specifically, the graph theory descriptors of the attention matrices reached the highest values of the correlation coefficient, while the correlation with the results from the noise analysis was lower, especially for the betweenness.

### 3.4. Network Properties of the Input Attention Distinguishing Gene Regulatory Network Architectures

To better capture information from the attention matrices, we interpreted these as a graph and used local parameters from graph theory to quantify the role of each node (gene) in the complexity of the interaction network with other nodes (genes). To this end, we considered three local descriptors that are able to determine both local topological properties and network properties linked to the definition of the shortest path (see Methods). Using the matrices obtained for the clustering coefficient, the hub score and the betweenness centrality from the input attention matrix of each GRN, we performed hierarchical clustering, as detailed in the Methods section. From the dendrograms shown in Figure 5, we can observe a tendency in separating oscillating networks and networks controlled by an oscillating external signal from the others. This was more accentuated for the clustering coefficient, meaning that the local structural network properties, captured by the clustering coefficient as a measure of local interactions with first neighbors, kept more the information necessary to recognize GRNs with different architectures.

We ran a PCA on the same matrices and we represented the GRNs and the partition in clusters using the first two principal components (PCs), which, together, explained, on average, 72% of the variance, for each network property studied. For the clustering coefficient (Figure 5A), the “MasterRegulator” and the “SparseConnection” networks were widely separated from the others along PC1. Regarding the betweenness (Figure 5B), the oscillating network with $n_{r}=n_{a}=10$ stood out along PC2, while the other networks were arranged along a continuous trajectory. Finally, for the hub score (Figure 5C), the separation along PC1 was driven by the network with the external signal with $n_{r}=n_{a}=10$, as also reflected by the clustering shown on the right.

### 3.5. Differential Response to Noise in the Prediction of Time Series Gene Expression Data

In this section, we present the study of the matrices of the MSE obtained from the analysis of noise described in the Methods and in Section 3.3. An example is shown in Figure 6A for two gene regulatory networks. The rows contain genes, ranked according to the mean over the different noise levels of the MSE (shown in the last column). As expected, we observed an increase in the MSE of the prediction increasing the variance $\sigma^{2}$. However, the structure of the matrices was different for different gene regulatory network architectures; for an oscillating network controlled by four clock genes (genes 0, 1, 2 and 3), the prediction by the DA-RNN was very accurate and stable against the addition of noise for some genes, while, for others, the MSE was larger and it increased rapidly with $\sigma^{2}$. Notably, the top four genes were the clock genes mentioned above, although the same ranking was not observed for all the oscillating networks used in this study. For the network controlled by a master regulator, the structure of the MSE matrix was more uniform, showing less differences between the genes in responding to noise addition.

Next, we built a matrix containing the means and the variances of the MSE for each gene and each GRN and we performed hierarchical clustering and PCA on it, as described in detail in the Methods section. The dendrogram representing the hierarchical clustering is shown in Figure 6B. We observed that the oscillating networks and GRNs controlled by an oscillating external signal were clearly separated from the others. This is somewhat similar to what we observed from the network properties of the input attention, but even sharper. The “FullyRepressed” and “MasterRegulator” GRNs clearly stood out, since, in these networks, most of the genes randomly oscillated around zero. From Figure 6C,D we can notice that PC1 reflected the top branching of the dendrogram, with the “FullyRepressed” and “MasterRegulator” GRNs widely separated from the other networks, as also found in the clustering shown on the right, while PC2 showed a finer separation between the other GRN architectures.

### 3.6. Comparison between Clustering of Gene Regulatory Networks

Finally, we compared the dendrograms obtained from the hierarchical clustering for the different network properties and for the analysis of the noise by computing the information-based generalized Robinson–Foulds distance, or tree distance (see the Methods section). The results are shown in Figure 7A. The most similar dendrograms were those obtained for the hub score and betweenness, while the farthest were those obtained for the betweenness and noise analysis, in agreement with the comparison of the matrices shown in Figure 4. The result given by the betweenness and hub score comparison is not surprising, because both descriptors measure the degree of centrality of a node in the complexity of the network. Nonetheless, they are two different types of centrality. The centrality of the betweenness is expressed in terms of shortest paths, while the hub score centrality is referred to the eigenvalues spectrum of the adjacency matrix. More interesting is the result concerning the difference between the clustering obtained with the betweenness and that obtained with the noise analysis. These two clustering analyses provided the most distant results and this shows how these two corresponding descriptors captured different properties of the system, which influenced the final results. We also compared the partitions obtained by cutting the dendrograms according to the silhouette analysis using the variation of information (see the Methods section for details). The most similar partitions were those obtained from the noise analysis and for the clustering coefficient. Indeed, we had already noticed that those matrices led to a better separation between oscillating networks and the others.

## 4. Discussion

In this manuscript, we used a DA-RNN to predict the behavior of stochastic processes. The aim of the work is not only to recover the time traces of the system, but also to infer information on its structure. A central point of this study is that we used classical models of gene regulatory networks to define a stochastic system for the simulation of our case studies. Using the in silico data generation approach, we altered the internal parameters of the system, whose behavior was then reproduced through the DA-RNN. This approach gave us full control over the interpretation of the results, preparing the ground for future applications of the outputs to real experiments. Indeed, state-of-the-art techniques for the measurement of gene expression, such as RNA-seq, can only provide time traces of gene expression, hiding the relationships among the components of the system. The inference of interactions among genes is a critical goal for biology, since understanding the effect of a gene on another allows one to predict the behavior of a gene regulatory network upon perturbations, opening the possibility to design new therapies to reshape the network.

Our work shows that the DA-RNN accurately reconstructed the time traces of genes belonging to different types of gene regulatory networks. In particular, we generated synthetic data from networks with different degrees of connectivity (from fully connected to sparser networks), networks displaying oscillatory behavior or driven by an external signal and networks with a master regulator gene. However, looking at the internal parameters of the attention layer of the neural network, we could not fully reconstruct the internal connection among the genes. Using tools from graph theory, we went beyond this lack of interpretability of the neural network parameters and we showed that considering the network properties of the input attention matrices, such as the clustering coefficient, the betweenness centrality and the hub score, it was possible to obtain information about the type of gene regulatory network under study. In particular, the clustering analysis showed that these network properties allowed one to distinguish different GRN architectures, with the clustering coefficient reflecting this structure better than other properties. We also studied the change in accuracy of the prediction by the neural network under noise addition on protein concentration. Performing a similar analysis, we showed that the response to noise of GRNs allowed one to separate different GRN architectures. Moreover, this analysis suggests that the core oscillating genes in oscillatory GRNs were more robust to noise addition, from the point of view of the ability of the neural network to predict their time traces, compared to the others, while, for a network controlled by a master regulator, all the genes responded in a similar way.

In this work, to train the DA-RNN, we used all the protein time traces but only one sample for each network. We already obtained a very accurate prediction of the system dynamics, although the performance might be improved by considering multiple samples, possibly generated with different initial conditions.

An interesting application of our method is the analysis of time series produced by high-throughput experimental techniques, such as microarray or RNA-seq data [31,32,33,34,35,64,65,66]. Given the time series of a gene from such datasets, our work shows that it should be possible to understand to which type of gene regulatory network it belongs, from the properties of the input attention of the DA-RNN used for its prediction. This could be extremely relevant to retrieve information on the biological process in which the gene is involved. Moreover, the large-scale application of our method to all the genes in a dataset could provide a new method, able to predict the future gene dynamic but also to infer regulatory modules. This work is also different from previous works that used Hopfield neural networks with a symmetric connectivity matrix to model the GRN dynamics storing the observed RNA-seq data as stationary states [67,68], because we actually constructed the dynamical model in order to predict the actual dynamics of the system and we used this model to classify different network structures.

In conclusion, we here propose an analysis of gene regulatory networks based on deep neural networks, which exploits the recently introduced DA mechanism and the power of RNN for time series prediction. We were able to reconstruct the behavior of the system with high accuracy for most of the GRN architectures. Moreover, through a network analysis of the internal parameters of the input attention, we could discriminate the different classes of gene regulatory networks, overcoming the lack of a direct connection between the internal parameters of the DNN and the physical quantities that describe gene interactions. Summing up, our work paves the way to the development of a method for simultaneous time series prediction and analysis of gene interactions from real gene expression data.

## Figures and Tables

**Figure 1 entropy-24-00141-f001:**
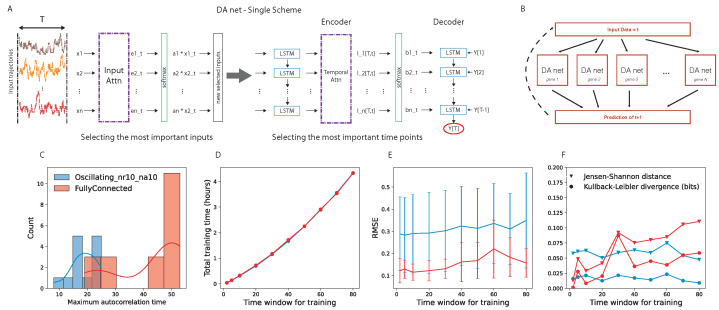
(**A**): Scheme of the dual attention recurrent neural network (DA-RNN) used in this study. From left to right: The network was trained on gene expression time traces of length *T*. The first attention layer (input attention) encoded which input time traces were more important for target prediction. The second attention layer (temporal attention) encoded the most important time points of the input time traces for the prediction. The LSTM chain on the right was the decoder that provided the final output. (**B**): Architecture of the parallel DA-RNN. Each box represents a DA-RNN devoted to the prediction of gene *i*; after the prediction of the next time step, the data were collected to fuel the inputs for the parallel nets and to predict the next time point. (**C**): Distribution of the autocorrelation time for the genes of a fully connected and an oscillating network (see Section 3.2 for details on different GRN architectures). (**D**): Training time for all the genes as a function of the time window *T* used for training, for the same two networks. (**E**): RMSE as a function of the time window *T* used for training, for the same two networks. (**F**): Kullback–Leibler divergence (circles) and Jensen–Shannon distance (triangles) between the probability distribution of the real data and that of the predicted ones, plotted as a function of the training window, for the same two networks. The values shown for each value of the time window are averages over all the genes of the GRN.

**Figure 2 entropy-24-00141-f002:**
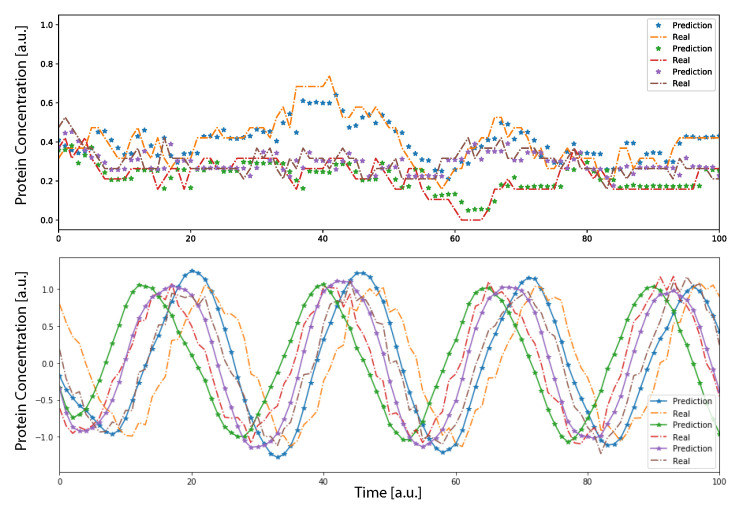
Gene regulatory network dynamic with N=3 genes. The dynamic was generated using a random interaction matrix $J_{ij}$. The top panel shows the continuous time trace of proteins concentration for a random GRN with $n_{r}=n_{a}=0.2$. The bottom panel shows the oscillatory dynamic of 3 genes relative to the core clock of an oscillatory network. In both panels, dots show the prediction of the neural network implemented using the first T=10 time points and then letting the system evolve on its own. The stochastic data were generated using the Gillespie algorithm.

**Figure 3 entropy-24-00141-f003:**
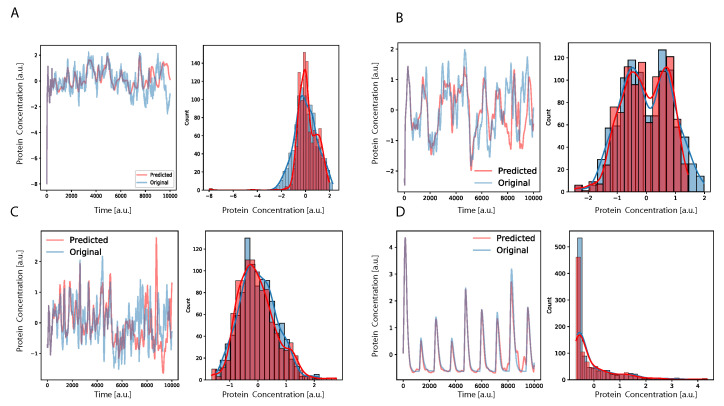
Time dynamics and full probability function (integrated over all the time of the simulation) of having a protein at a certain concentration. We show selected example genes from an oscillatory GRN (panels **B**,**D**) and a GRN controlled by a master regulator gene (panels **A**,**C**), both with $n_{r}=n_{a}=0.5$. The plots show the performance of the prediction comparing the time traces between the stochastic simulation and the DNN propagation. Moreover, the resulting probability distribution of the given protein concentration for all times is reported. We highlight that the protein concentration dynamic was reconstructed with high accuracy and even the noise amplitude and the mean were quite well reported by the neural net. Moreover, for the genes belonging to the oscillating GRN (panels **B**,**D**), it was possible to reconstruct the amplitude and the mean of the oscillations. We stress that the dynamic shown in panel **B** could not be immediately addressed to a gene driven by an oscillatory clock, but the bi-modal distribution shows that this was the case. Interestingly, this reveals that, given a model trained on real data, we can look at its internal parameters that allow us to determine if the genes in the GRN belong or not to a certain network topology such as an oscillatory-driven network, such as, for instance, the cell cycle or circadian clock.

**Figure 4 entropy-24-00141-f004:**
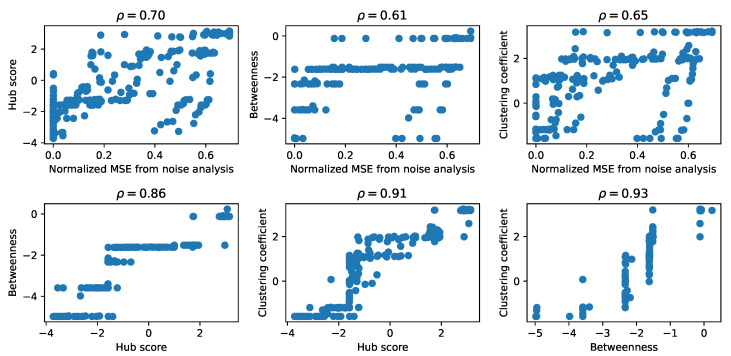
Comparison of the matrices resulting from the input attention and response-to-noise analyses. Scatter plots showing the relationship between the matrix elements of those obtained computing the network properties of the input attention (clustering coefficient, hub score and betweenness) and the matrix of normalized mean MSE obtained from the analysis of the response of the prediction to noise. Note that the data are represented in log–log scale using the transformation sign(x)ln(1+|x|), since they included both positive and negative values. The Pearson correlation coefficient ρ between each pair of properties, computed on the transformed data, is reported on top of each panel.

**Figure 5 entropy-24-00141-f005:**
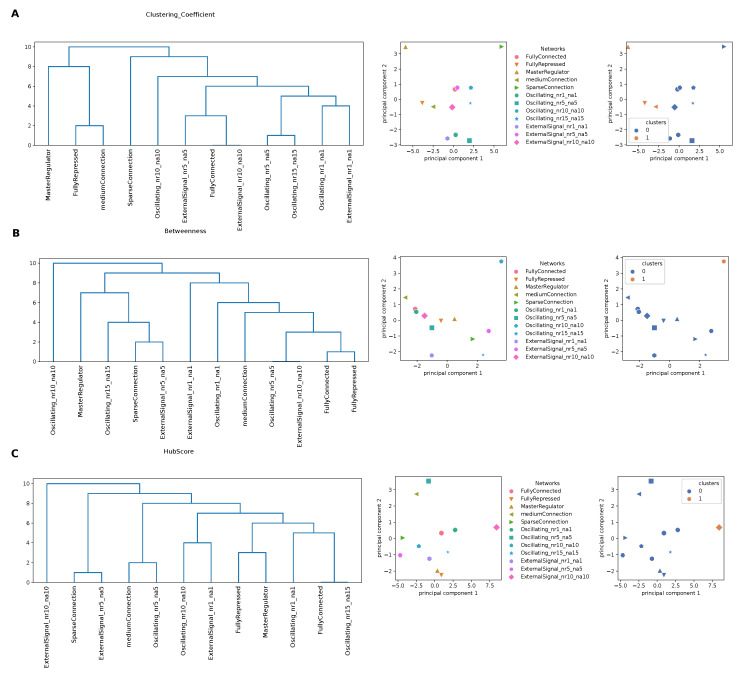
Clustering based on the network properties of the input attention matrices. We show the dendrograms obtained from a hierarchical clustering (**left**), a PCA plot showing the 12 gene regulatory networks used (**centre**) and the obtained partition in groups (**right**) for the clustering coefficient (**A**), betweenness (**B**) and hub score (**C**).

**Figure 6 entropy-24-00141-f006:**
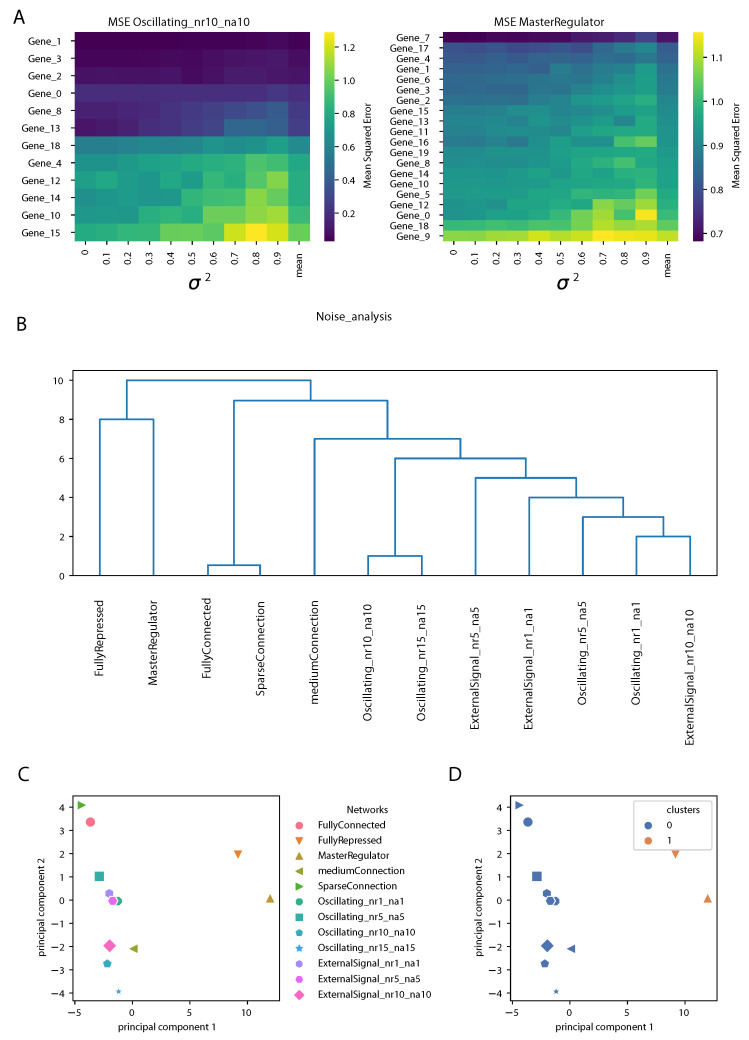
Impact of noise addition on time series gene expression prediction for different GRN architectures. (**A**): Matrices of the mean square error (MSE) on the prediction by the DA-RNN, for an oscillating GRN with 10 repressors and activators per gene on average (**left**) and a GRN controlled by a master regulator (**right**). Rows (genes) are ranked according to the mean of the MSE (last column). (**B**): Dendrogram obtained from the hierarchical clustering of the matrix summarizing the response of the prediction to noise for each gene regulatory network. (**C**,**D**): PCA showing the 12 GRNs used and their partition in clusters.

**Figure 7 entropy-24-00141-f007:**
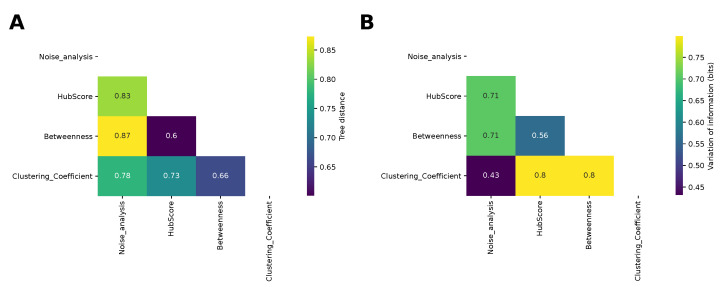
Comparison between the clusterings. (**A**): Comparison between the dendrograms of the different hierarchical clustering shown in Figure 5 and Figure 6 using the information-based generalized Robinson–Foulds distance (or tree distance). (**B**): Comparison between the partitions obtained using the variation of information.

**Table 1 entropy-24-00141-t001:** Prediction accuracy of the DA-RNN for different gene regulatory network architectures. The network types are those described in Section 3.2, where $n_{r}$ and $n_{a}$ indicate the mean number of repressors and activators per gene, respectively. We report the mean and standard deviation of the root-mean-square error (RMSE) computed over all the genes in each network.

Network	Mean (RMSE)	Std (RMSE)
FullyConnected	0.12	0.04
FullyRepressed	0.67	0.08
MasterRegulator	0.83	0.04
SparseConnection	0.14	0.05
mediumConnection	0.35	0.23
Oscillating_nr1_na1	0.22	0.11
Oscillating_nr5_na5	0.23	0.13
Oscillating_nr10_na10	0.30	0.17
Oscillating_nr15_na15	0.12	0.09
ExternalSignal_nr1_na1	0.29	0.20
ExternalSignal_nr5_na5	0.39	0.28
ExternalSignal_nr10_na10	0.09	0.06

**Table 2 entropy-24-00141-t002:** Pearson correlation coefficient ρ between input attention and true interaction matrix.

Network	ρ
FullyConnected	−0.018
FullyRepressed	0.0
MasterRegulator	−0.080
SparseConnection	−0.10
mediumConnection	−0.0087
Oscillating_nr1_na1	−0.0070
Oscillating_nr5_na5	0.017
Oscillating_nr10_na10	0.061
Oscillating_nr15_na15	0.12
ExternalSignal_nr1_na1	−0.0020
ExternalSignal_nr5_na5	0.075
ExternalSignal_nr10_na10	0.021

## Data Availability

All the data and code needed to reproduce the analysis presented in the manuscript are available at the Github repository https://github.com/jonathan-f/DA_RNN_GENEXP, accessed on 12 December 2021.

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
