# Peer review of "Prediction of Time Series Gene Expression and Structural Analysis of Gene Regulatory Networks Using Recurrent Neural Networks"

_entropy, 2022, doi:10.3390/e24020141_

Round 1
Reviewer 1 Report
With this revision, I find the manuscript much better in term of readability. However, there are still issues in motivations and confusions. Also, as this is strictly a method paper, I would recommend providing and describing the software code with the submission.
Major comments:
-General: The paper focus shifts often between the efficiency of DA-RNN and the role of different network architectures. Thus, it would need a comparison of DA-RNN inference performance to other methods (e.g. Bayesian models). This of course requires major work. Alternatively, the authors should discuss more the uniqueness of DA-RNN: the prediction capacity of DA-RNN and the information stored in the attention layer (matrix A).
-Discussion regarding the training window should be put in a separate section, with proper mention to the simulation scheme:
-I find the sudden mentioning of T=50 (line 187) without the context very perplexing. Maybe the author should discuss the motivation for testing different training window first, or move the paragraph to after line 374.
-Line 205: To train …, we use one time series with 1001 time points (sampled every 10). This is confusing since the training window mentioned above is T=50 (i.e. 5 time points per sample?).
-Line 206, the mentioning of “sampled every 10” is alerting. Are time (a.u.) and time step different? What is the unit of T, in time steps (as in Line 236) or absolute (as in Line 187)? Please unify and clarify. Also in Fig. 2, the sampling time is every 1 time unit, not 10.
-Line 357-361, in light of the performance test with different time window T, maybe the statements can be removed.
-Line 437-line 448: added Gaussian noise to the signal after the simulation should only be interpreted as error in measurement, e.g. in Live-Seq experiment, only a subset of the cytoplasmic content is measured. I find the mentioning of mutations and other sources of variability in gene expression and regulation, which are intrinsic noise (in answer to old question 9), as motivation for this section is not proper.
-In Figure 4, the motivation to study the correlation between learned network features is still not clear to me. The learned features are a combination of 1) true network wiring and 2) the learning model (in this case, DA-RNN) and 3) noise.
-Correlation does not necessarily come from the true network wiring but can come from the latters (2 and 3) as systematic error/bias. Please discuss this.
-What the feature score learned directly from true wiring matrix Jij? How is it different from the learned wiring matrix A with zero noise and with increasing noise?
-The inference is currently done on a single time trace. Is there any possibility to improve the accuracy of prediction or learned features with more independent samples (i.e. more cells, more replicates)? Please discuss in Discussion.
Minor comments:
-Line 163: what is
-Figure 3. Please add x axis’s label
-Table 1. Please state many replicates were made to obtain the standard deviation of RMSE
-Figure 3. To avoid confusion about the model starting state, please remove the initial time points of the predicted traces (which can take drastic values of -8 in A) and readjust the y-axis. This will make evaluating the prediction accuracy easier.
Author Response
Thank you very much for your comments, please find attached a word doc with the answers to your questions.
all the best
in behalf of the Authors
Michele Monti and Jonathan Fiorentino

Reviewer 2 Report
In my opinion this is an interesting and well written paper. I have following minor comments:
1. There should be a sepparate section on data that has been used in experiment. Is data artificially generated (Appendix A)? Why? Please discuss how artificially generated data affects results.
2. Figure 6 B - there are 12 captions but only 11 tree branches - is this an error?
3. As far as I understand results have been caluclated in Python. In my opinion authors should publish data and source codes for readers in order to reproduce the reserch.
Author Response
Thank you very much for your comments, as you asked we have created a GitHub repository with the codes. Moreover in the attached file you can find detailed answers to all the reviewers comments.
all the best
on behalf of the Authors
Michele Monti and Jonathan Fiorentino

This manuscript is a resubmission of an earlier submission. The following is a list of the peer review reports and author responses from that submission.
Round 1
Reviewer 1 Report
This paper addresses the gene regulatory network problem by employing Deep Neural Networks. Introduction Section is nicely written and provides all the relevant information. The materials and methods section contains the details of proposed approach along with the case study to validate the proposal. Results are then nicely presented in Section 3. My only concern with this paper is the performance comparison. Authors are required to compare their results with recent state-of-art solutions, using tables, graphs and any other methods used for the comparison. Good data visualization techniques may enhance the value of their article. In this context, authors can use various performance parameters for comparisons. Each performance parameter must be defined explicitly before its use. Authors should also provide the justification and motivation behind the selection of each performance parameter.

Reviewer 2 Report
The manuscript by Monti et al. aims to showcase the power of the recurrent neural network (RNN), developed from neural network for learning time-series data. The technique is demonstrated on stochastic simulation data of gene regulatory network (GRN) and showed that the learned network retains key features of original network.
The manuscript is well written and easy to access. The topic of implementing deep learning techniques deservedly in research generated lots of attention due to their prediction capacity. However, to utilize them as central tools in knowledge extraction remains challenging due to the lack of statistical power for model selection. The manuscript is thus welcomed as an attempt to evaluate which information from the learned model can be useful.
However, the work’s major pitfall is in the unrealistic data assumption: the temporal expression dynamics of multiple genes. This is impossible with high-throughput techniques (sequencing, FISH…), which requires sample fixing. Live imaging with fluorescent-tagged proteins or RNAs allows to measure at max 3-5 gene dynamics due to fluorescent channel interference. Thus, I find the relevance/significance of the results beyond the small gene network minimal.
Also, many technical aspects of the simulation and inference, regarding e.g. simulation parameters, number of samples, the learning time window… are not properly discussed. Therefore, I am not able to judge the applicability of the work to study realistic gene networks.
Major comments:
- The introduction can benefit from discussing more literature on the rich field of computational biology in studying network architectures and some attempts to use deep learning in knowledge extraction in biology.
- What realistic measurements are intended for the high-throughput monitoring of gene expression dynamics as generated by the Gillespie algorithm?
- What are the simulation parameters used in modeling 3 gene repressilator, the large gene networks? , , (Eq. 9). How is generated/randomized?
How many samples and time points are required for the learning of gene network?
What is the sampling time?
- In Fig. 9, how can the delay term be simulated in Gillespie algorithm? The term implies the system needs to know the future?
- In the proposed model of gene expression regulation, transcription is not modeled explicitly (rather considered a delay). Therefore, the protein number is expected to be Poisson-like distributed instead of gamma-like (Taniguchi et al, Science, 2010, Pedraza & Paulsson, 2008). When the protein number is high, intrinsic noise is low and the system become deterministic (close to mean-field solution). This low intrinsic noise explains why the predictions for the future traces as in Fig. 1 can look good at all. It is not clear how the prediction fares when noise is higher.
- The simulation, as described in section 1.1. is pure Markov, with no hidden state. The system evolution depends only the current state but not the past. Why there is a need to use a memory-based learning network to study a Markov model? Would a simple RNN perform as good?
- Line 336 - 347: How learning time window is defined for the oscillator traces? As the signal is periodic, the autocorrelation function should have multiple peaks corresponding to multiples of the oscillating period. How the maximum autocorrelation time is extracted?
As stated by the authors, this time window has a fundamental role. Therefore, the choice of the time window needs clarifying (e.g. performance does not increase significantly beyond specific window length, or comparing with brute force approach).
The work in Tostenvin et al is applied for traces with Gaussian-variation around at steady-state, where the “autocorrelation time” is close to the effective sampling time. Thus, I don’t find the relevance to periodic signals.
- I don’t understand Fig 3 and line 411-414. Please clarify and provide intuitions.
Also, how robust is the inference of the different architecture features (hubscore, betweenness, clustering) to adding noise.
- Regarding noise addition:
When gaussian noise is added? Is this done during or after the Gillespie simulation?
Line 226: what is “the previous time step of the target gene”? Again, sampling time here is important.
Personally, I find this idea really awkward due to an artificial continuous noise added on top of the discrete stochastic protein counts. This is a hybrid of PDE and SSA (Gillespie), while the SSA by itself generate tunable noise by e.g. adding an intermediate transcription (see Pedreza & Paulsson, 2008; Ribeiro 2007).
Minor:
- In Fig. 1, why is the protein concentration (a.u) shown instead of number (typical Gillespie algorithm output)? Without the number, it is difficult to judge the impact of intrinsic noise here.
- Line 475: Betweenwss
- The whole 2.1 section about Gillespie Algorithm is quite standard and can be omitted for manuscript clarity.
- Eq A1. Should it be ? or is negative?
- What is hubscore, Betweenness, And Transitivity? And the intuition when interpreting these parameters? These are all relative foreign network concept especially for biologists.
- A figure on the DA-RNN structure would be helpful for non-data scientists to understand better the learning scheme.
- Line 339: as each time point is considered an independent sample, why a big time window would provide confusing information to the network?
- In Fig. 3, the top panels axes should be swapped for study the effect of different network architecture.